

# Design of public cultural sign based on Faster-R-CNN and its application in urban visual communication

Xiang Gong and Jingyi Fang

Department of Arts Design, Visual Communication Design Major, Dankook University, Yongin-si, Korea

## ABSTRACT

People are increasingly enthusiastic about pursuing spiritual life as economic and social development continues. Consequently, public cultural content has emerged as a pivotal instrument for promoting international soft power across diverse nations and regions. In today's era of advanced artificial intelligence, cultural sign design optimization has become achievable through its deployment. This article establishes an automatic layout optimization framework, specifically tailored to meet the visual communication requirements of public cultural signage. Our framework employs Faster-R-CNN for detecting and extracting key elements of the poster, yielding an impressive average detection accuracy of 94.6%. Subsequently, we use the three-division method in design to optimize the layout, ensuring that cultural logo design conforms to visual communication principles. Our framework produced an average cultural logo satisfaction rating exceeding 70% in actual tests, providing novel insights for cultural sign design within the artificial intelligence context and significantly enhancing the efficacy of visual communication conveyed through such signage.

## INTRODUCTION

Public cultural signage represents an amalgamation of public and cultural signs that exist in public spaces and pertain to cultural services and amenities. As a grassroots initiative to construct public cultural facilities, public signs serve as visualized displays of cultural services offered by public cultural institutions, and also help in enhancing their image and expanding their reach (*Angus, 2022*). While there have been significant achievements in the research, design, and application of public signs across various disciplines, including architecture, system engineering, psychology, and ergonomics, the theoretical basis is intricate and the academic community is yet to reach a consensus (*Boniotti, 2023*). The primary functions of public cultural signage are to promote public cultural institutions, enhance the cultural image of cities, improve users' emotional satisfaction, boost the utilization rate of public cultural facilities, and ensure standardization of cultural services. Therefore, there is a pressing need to enhance the efficiency of visual communication, in order to improve the design process and image of cultural signs (*Pan, 2021*).

Corresponding author
Xiang Gong, gxsdtz@126.com

Visual communication employs visual symbols to convey a diverse range of messages. It represents a form of communication between the sender and the receiver, using visual perception and language to express communication (*Sless, 2019*). Communicators and receivers belonging to different regions, genders, ages, and using diverse languages, interact with each other through visuals and media (*Bannister et al., 2021*). Visual communication plays a crucial role in the design of logos, and by incorporating varied ways of thinking and expressions through visual communication methods, relevant visual elements can enhance the efficacy of communication.

In the design process of cultural identity, its communication is mainly done through posters, and the designed cultural image is embedded in the posters to complete its communication (*Hou et al., 2019*). At this time, it is necessary to optimize its layout design, and the graphic layout has similarities with the layout of poster design, especially the layout design of magazine covers, through the combination and arrangement of limited visual elements to achieve the purpose of visual attraction and information conveyance, which coincides with the function of posters. The automatic graphic layout system proposed by some scholars is not limited to magazine media, but other media including books, posters, slides, *etc.* are also applicable (*Liu et al., 2019*). The three core points of layout design are design elements, design principles and design aesthetics, so it is important to extract and separate these elements to achieve automatic optimization of layout for the design and communication of cultural identity. With the development of artificial intelligence technology, the use of neural network and image processing technology to identify the symbols in cultural logos, separate the core elements in the layout, and match them with the classical proportions in visual communication makes intelligent layout optimization possible. The benefits of AI typesetting and target recognition technology in layout design are mainly reflected in the following two aspects:Firstly, artificial intelligence and intelligent layout technology can automate the task of layout design, improve design efficiency, and save designers' time and energy. Intelligent typesetting technology can automatically adjust the position, size, font, and line spacing of elements such as text and pictures, while ensuring the overall balance and aesthetics of the layout and improving the design effect. Secondly, target recognition technology can help designers more accurately identify and select appropriate elements such as images, colors, and styles for layout design, improving the visual effect of the layout and the accuracy of conveying information. Through target recognition technology, designers can quickly find images and color combinations related to the theme of the layout, improving the attractiveness and readability of the layout.

Therefore, this article uses deep neural networks in artificial intelligence to identify and segment the relevant elements in the posters as objects, and converts the design problem into a target matching problem through the conversion of design logic and computational logic to complete the visual optimization so as to realize the design of public cultural signs based on artificial intelligence methods, and contributions of this article are as follows.

1. According to the communication needs of public signs and based on the visual communication background, this article proposes a modular sign design form.

2. Using artificial intelligence technology, the module monitoring of the three-element poster was completed by Faster-R-CNN network, and its detection rate was 94.6%.

3. According to the method of three divisions in visual communication, the elements detected by image segmentation were used for layout optimization and reformatting, and the satisfaction rate of the new logo obtained exceeded 70%.

The remainder of this investigation is organized as follows: 'Related Works' investigates related works for deep learning and object detection in the layout design; In 'Image Segmentation and Layout Composition Modeling', the Faster-R-CNN used for the model establishment is introduced; 'Experiment Results and Analysis' describes the experiment and result analysis of the element. In the 'Discussion', we discuss the result and what can AI do for the layout optimization in the future; the conclusion is presented in 'Conclusion'.

# RELATED WORKS

## Deep learning and image segmentation

The first task of layout design using artificial intelligence is to make the computer understand the layout construction and elements. Deep learning is a feature learning, where features are learned from the data in the training set, and is essentially a multi-level nonlinear combinatorial learning method. The development of deep learning is based on statistics and machine learning, and one of the first neural networks to be proposed was the perceptron. Deep learning has shone in many fields, and the field of computer vision is one of them. Deep learning based algorithms have achieved significant result in the image analysis, detection, localization, segmentation, *etc*. In the field of image segmentation, there were early methods based on thresholding, boundaries, and region growing, which relied on a priori knowledge and required manual design of features with cross-sectional effects. In 2015, the emergence of FCNs led researchers to turn to fully convolutional neural networks, and many of the networks that emerged later for image segmentation evolved based on FCNs. For example, models such as SegNet (*Badrinarayanan, Handa & Cipolla, 2015*), DeepLab (*Chen et al., 2017*) and RefineNet (*Lin et al., 2017*) andDANet (*Fu et al., 2019*) have been proposed by researchers and have made a splash in the field of image segmentation. Object detection and target recognition in nature are both more popular image segmentation techniques, such as damage monitoring, *Cha, Choi & Büyüköztürk (2017)* pioneered the application of convolutional neural networks to concrete crack detection, using the establishment of a five-layer convolutional neural network and using a sliding window to process larger images with better edge detection compared to traditional edge detection methods such as the traditional Canny operator and Sobel operator. performance. AlexNet (*Maeda, Sekimoto & Seto, 2016*) classification network was used to classify road pavement images, but the network was only able to locate whether there was a defective part in an image and could not locate the location of cracks in the pavement. *Maeda et al. (2018)* used SSD target detection model for detection and identification of road pavement damage and could find the approximate location of the defective part in the pavement image. In addition, image segmentation has a wide range of applications in the medical field, and in 2019 *Ibtehaz & Rahman (2020)* proposed the Multi-ResNet network after borrowing from ResNet to fuse the residual module and modify the convolutional blocks and jump connections of the U-Net network. In 2020, *Huang et al.*

*(2020)* proposed the U-Net3+ network model, which uses full scale jump connections and deep supervision, proposed a hybrid loss function, added a classification bootstrap module to address over-segmentation of non-organic images, and therefore achieved more accurate segmentation results. *Li et al. (2020)* proposed the ANUNet network model based on the UNet++ network model, embedded an attention mechanism in its jump connections, and designed a hybrid loss function is used to solve the data imbalance problem.

The extraction of objects to be detected in the target image through image segmentation techniques to accomplish the focus on relevant information is the current research focus. As can be seen from the above related studies, these studies improve the recognition accuracy and segmentation accuracy by improving the traditional or advanced neural networks to ensure that they achieve the desired goals. Therefore, the objective can be achieved by selecting the appropriate model according to the data characteristics.

## Target detection and automatic layout design

For public signage and poster design, the separation of the target is completed through image segmentation technology, while converting into a target matching pair to optimize the layout of the layout, which can ultimately achieve the optimization of the target layout. Through the combination and arrangement of limited visual elements, the purpose of visual attraction and information communication can be achieved, which coincides with the function of posters. The automatic graphic layout system proposed by some scholars is not limited to magazine media, but other media including books, posters, slides, *etc.* are also applicable. Therefore, the investigation of the automatic layout generation system for magazines proposed by foreign scholars will help this article to study the intelligent generation of poster layout.

*Jahanian (2016)* have studied several excellent image text layouts and summarized three core points of layout design, which are design elements, design principles and design aesthetics here design elements refer to the six basic visual elements of color, line, shape, style, texture and volume; *Odonovan, Agarwala & Hertzmann (2014)* proposed an automatic layout system for single-page graphic design works, which is designed by significantly region calculation, alignment detection, and grid segmentation to optimize the layout. *Yang et al. (2016)* argue that good layout design should consider domain-specific aesthetic principles and computational content features related to the subject matter before using predefined templates. Therefore, they proposed an automatic image text layout generation system that contains a set of subject templates designed based on expert experience and a computational framework based on energy optimization. *Li et al. (2019)* proposed a layoutGAN network to synthesize layouts by modeling the geometric relationships of different types of 2D elements The network is mainly used to study the stacking relationships of image elements and generate graphical with correlated layouts. However, *Tabata et al. (2019)* argued that layout-GAN layouts are generated from known cumulative feature distributions, which makes it difficult to create novel layouts, and propose a system that uses a minimal conditional rule set randomization process to generate layouts, which first extracts the original design layout and then randomly generates candidates using the minimal conditional rule set to achieve layout diversity. *Goodfellow*

*et al. (2020)* proposed the Generate Counter Network (GAN) model, which consists of a generator and a discriminator and can generate fake data similar to real data and perform image generation and conversion.

According to the review of current research, it can be found that the current automatic layout design technology is mainly done through image detection and matching, which is similar to the idea of image segmentation. Therefore, through the current more mainstream target detection algorithm, the main body of the designed logo is detected and separated, and the relevant elements are used to match, so as to achieve the optimization of visual carriers such as posters and improve the efficiency of communication.

## IMAGE SEGMENTATION AND LAYOUT COMPOSITION MODELING

### The marker detection in signs design using Faster-R-CNN

According to the image detection and segmentation techniques presented in section2, deep convolutional networks have an advantage in such studies. Faster-RCNN is developed and improved from R-CNN, Fast-RCNN, and has gradually become a classical and representative algorithm in target detection (*Maity, Banerjee & Chaudhuri, 2021*). Faster R-CNN is a popular object detection algorithm based on convolutional neural networks (CNNs) which combines a region proposal network (RPN) with a Fast R-CNN object detection network. It works by first generating a set of object proposals, then refining and classifying these proposals using a CNN. The recognition process in Faster R-CNN involves two main stages. First, the RPN generates a set of object proposals by scanning an image with a set of pre-defined anchor boxes at multiple scales and aspect ratios. The RPN generates a set of scores for each anchor box, which are used to rank and select a subset of high-quality object proposals. In the second stage, the object proposals are passed through a Fast R-CNN network for refinement and classification. The Fast R-CNN network generates a set of scores for each class label, and uses a softmax function to output the final class probabilities for each proposal.The segmentation process in Faster R-CNN involves the use of a CNN-based segmentation network, which is applied to each object proposal generated by the RPN. The segmentation network generates a pixel-wise mask for each proposal, indicating the location of the object within the proposal. This allows for precise localization and segmentation of objects within an image.The operation algorithm in Faster R-CNN is designed to optimize both the region proposal and object detection stages. The RPN is trained to generate high-quality object proposals by minimizing the binary cross-entropy loss between the predicted objectness scores and ground-truth labels. The Fast R-CNN network is trained using multi-task loss, which combines the softmax classification loss and bounding box regression loss. The segmentation network is trained using pixel-wise binary cross-entropy loss, which penalizes incorrect pixel predictions (*He & Zhang, 2019*). Similar to the traditional CNN method, its main network structure is composed of several parts: input, convolution, pooling and classification, the most obvious of which is the convolution operation, whose formula is shown in Eq. (1).

$$Y^{b(x)} = \sum_d W^{db} c^{d(x)} + a^b \tag{1}$$

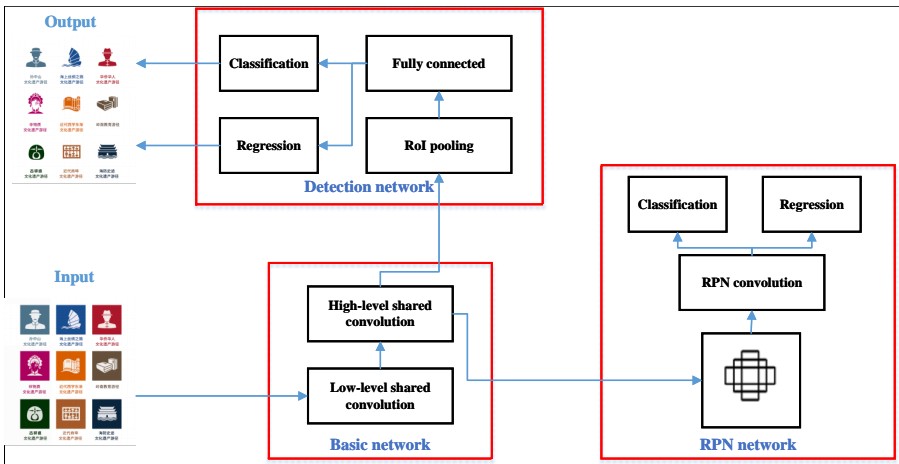

**Figure 1  Framework for Faster-R-CNN.** The Faster-RCNN architecture is shown, where the inputs are pre-processed and normalized, and then extract features and generate candidate frames using feature extraction network and region generation network, respectively.

where the position in the formula feature mapping is denoted by $x$ is denoted, $b$. The features extracted after the layer convolution operation are denoted by $Y^{b(x)}$ is denoted, $d$. The feature mapping before the layer convolution operation is denoted by $c^{d(x)}$ The deviation coefficients of the convolution kernel are denoted by $W^{db}$ and $a^{b}$ are denoted by and The convolution operation in the convolutional neural network makes the convolutional layers have the properties of local relations and weight sharing, which effectively avoid the problem of gradient explosion or disappearance. These two features is vital for reducing the number of parameters and making the operation simple and efficient when training large data sets.

Faster-RCNN is developed and improved from R-CNN and Fast-RCNN, and has gradually become a classical and representative algorithm in the field of general-purpose target detection. The Faster-RCNN architecture is shown in Fig. 1, where the inputs are pre-processed and normalized, and then extract features and generate candidate frames using feature extraction network and region generation network, respectively. The non-maximum suppression (NMS) algorithm is used on the convolutional shared feature map to remove the redundant prediction frames and find the best detection position. Next, the obtained individual candidate frames are mapped onto the feature layer to form candidate regions, which are pooled by the RoI pooling layer to obtain specific feature data. Finally, the fully connected layer is directly processed to output the probability values and the positions of the prediction frames.

The training process of Faster-R-CNN is divided into two stages, which are training RPN network and subsequent classification network, where there are two loss functions of RPN network are softmax loss and smooth L1 loss, which are calculated as shown in Eqs. (2) and (3) (*He & Zhang, 2019*).

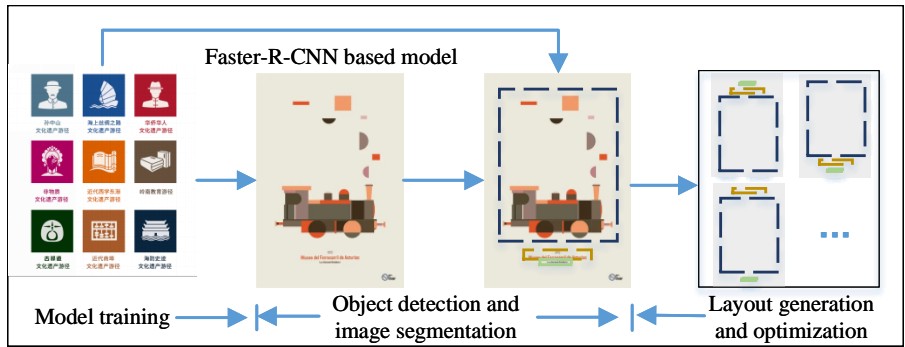

**Figure 2** **The framework for the intelligent layout generation.** These posters mainly consist of three parts: logo body, introduction and theme, and the background is relatively simple.

$$L\left(\{p_i\},\{t_i\}\right) = \frac{1}{N_{\text{cls}}}\sum_i L_{\text{cls}}\left(p_i,p_i^*\right) + \lambda\frac{1}{N_{\text{reg}}}\sum_i p_i^* L_{\text{reg}}\left(t_i,t_i^*\right) \tag{2}$$

$$L_{\text{reg}}\left(t_i,t_i^*\right) = \sum_{i\in x,y,w,h} \text{smooth}_{\text{L1}}\left(t_i - t_i^*\right) \tag{3}$$

where, for the smoothL1 function in Eq. (3), the specific expression is expressed in Eq. (4) as follows.

$$\text{soomth}_{\text{L1}}(x) = \begin{cases} 0.5x^2 & \text{if } |x| < 1 \\ |x| - 0.5 & \text{otherwise.} \end{cases} \tag{4}$$

After the extraction of cultural identifiers using Faster-R-CNN, these elements need to be rearranged to complete the final layout design, so the next section will describe how to perform the layout generation task. In this article, the ResNet-50 is used to constructed the network described in Fig. 1 with five convolutional layers.

## Faster-R-CNN based layout generation

The three main components of the Faster-RCNN target detection algorithm architecture are feature extraction network, candidate frame suggestion network and detection network. The operation logic of the matching model framework based on the layout composition method, firstly, requires the learner to learn the classification of layout elements using convolutional neural network, followed by migration learning of layout designs of different composition cases to form the initial template, then the generator optimizes the initial template using the golden ratio and trilateration parameters to produce multiple optimized templates. After that, the system matches the input design material category and quantity as well as the selected composition method to the corresponding template library, then automatically processes the material according to the template constraints, and finally produces multiple design solutions by adjusting the design optimization parameters. Different from the traditional classification task, the layout generation is based on the extracted original logo information for optimization to complete the generation of optimized templates. In this

article, the more classic rule of thirds is chosen, *i.e.,* by adjusting the alignment of elements, adjusting the position of the body layout, adjusting the spacing between the body and text elements, transforming the relative position of the body and text, and adjusting the position relationship between text elements.

The model training is also based on the analysis of the posters collected with these three types of information as the main body. In the process of optimization, only the spacing and visual effects of the layout are adjusted, so that the relationship between the main body, title and poster description is completed in the final design, in line with the rule of thirds, so as to achieve the optimal output, the overall framework is shown in Fig. 2.

# EXPERIMENT RESULTS AND ANALYSIS

The data set comes from https://research.google.com/youtube8m/index.html (Names: YouTube-8M Segments Dataset; Institutions: YouTube-8M Users Group). According to the poster layout generation framework illustrated in Section3, 134 posters are selected for training and testing of the model. These posters mainly consist of three parts: logo body, introduction and theme, and the background is relatively simple, similar to the example in Fig. 2. For the layout generation, the core focus is if the target object is monitored with high accuracy, *i.e.,* the learner needs to have high accuracy, so this article focuses on the monitoring efficiency of the model, and the results are analyzed as follows.

## Target detection and recognition rate

Considering deep learning and image detection and recognition, the evaluation model metrics are mainly done by Precision, Recall and F1score.

In the model building, we chose a structure with only 64 hidden nodes per layer due to the low complexity of the picture itself, and analyzed the model by comparing the characteristics of the loss function under different building layers. Figure 3 shows the loss function under 20 hidden layers and 50 hidden layers, and it can be seen that there is a decline and slow iteration under 50 layers.

Figure 4 gives the results of the evaluation indexes of the three different kinds of information.

For cultural identity communication, the accuracy of the detection target is intuitively important, so this article visualizes the Precision indicator for comparison, and the results are shown in Fig. 5.

It can be found from Fig. 5 that the Faster-R-CNN has significantly better detection accuracy than the CNN method and the R-CNN method, and the average detection rates of the three methods are 0.882, 0.900 and 0.946, respectively, which shows the superiority of the method used.

## Model test and application

The ultimate goal for the communication of cultural identity is to make it memorable. Optimizing them using artificial intelligence and investigating the satisfaction of the newly generated layout after the optimization is the most important for their future application. Therefore, in this article, relevant cultural signage content selected in the region was selected

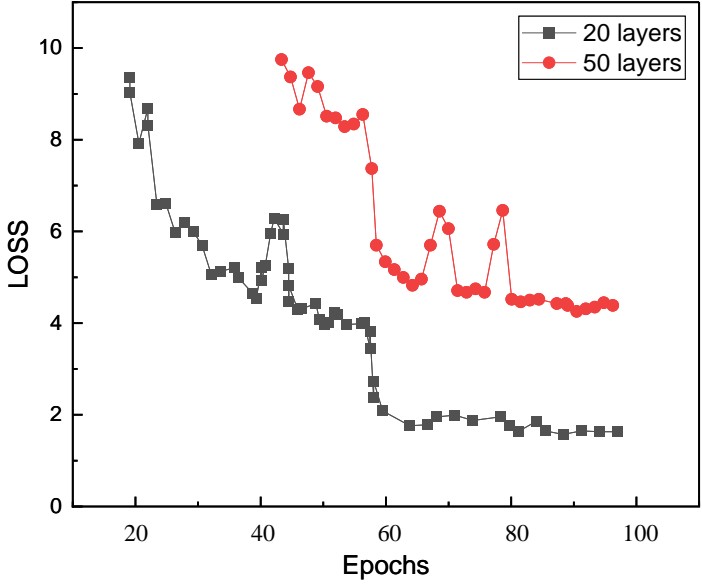

**Figure 3** **Model loss with different layers.** The figure shows loss function under 20 hidden layers and 50 hidden layers, and it can be seen that there is a decline and slow iteration under 50 layers.

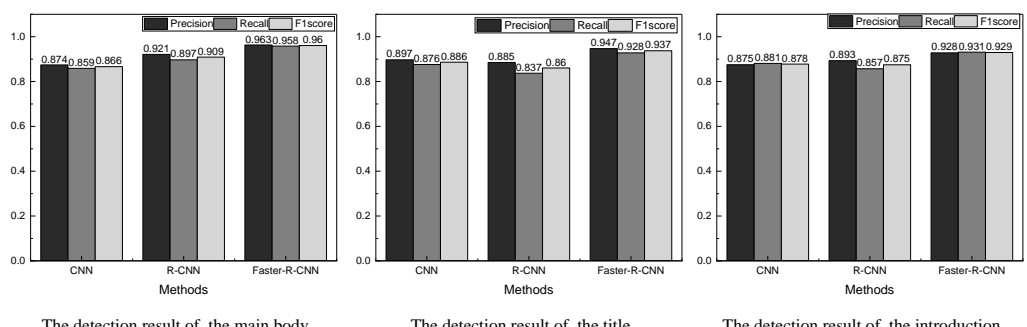

**Figure 4** **Results of the evaluation indexes of the three different kinds of information.**

for testing, and relevant volunteers were recruited to conduct pre- and post-optimization tests. In the actual test, two types of tests were conducted: Test 1 was an optimization that only adjusted the spacing and proportional relationship between the three elements, while Test 2 was a re-optimization of the element positions. Since the optimization of the element positions involved new permutations, adjustments were made during the tests based on the suggestions of the personnel who indicated the audit department. The results of the two types of tests are shown in Fig. 6.

In Fig. 6, we can find that for both types of tests, more than 70% of the volunteers think the adjusted posters have better visual communication, while those who think no adjustment is better or there is no difference between the two. For art design and communication, having a significant difference of more than 70% and the degree of liking
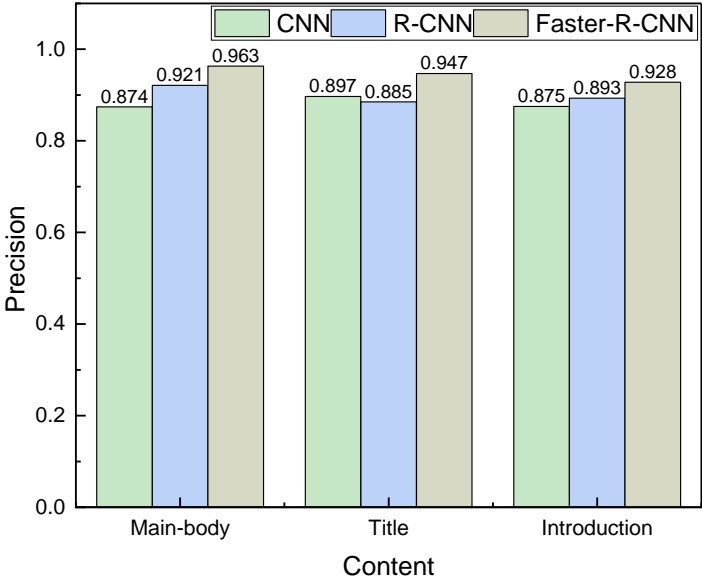

**Figure 5** **The precision of different objects detection among different methods.** The Faster-R-CNN has significantly better detection accuracy than the CNN method and the R-CNN method, and the average detection rates of the three methods are 0.882, 0.900 and 0.946, respectively, which shows the superiority of the method used.

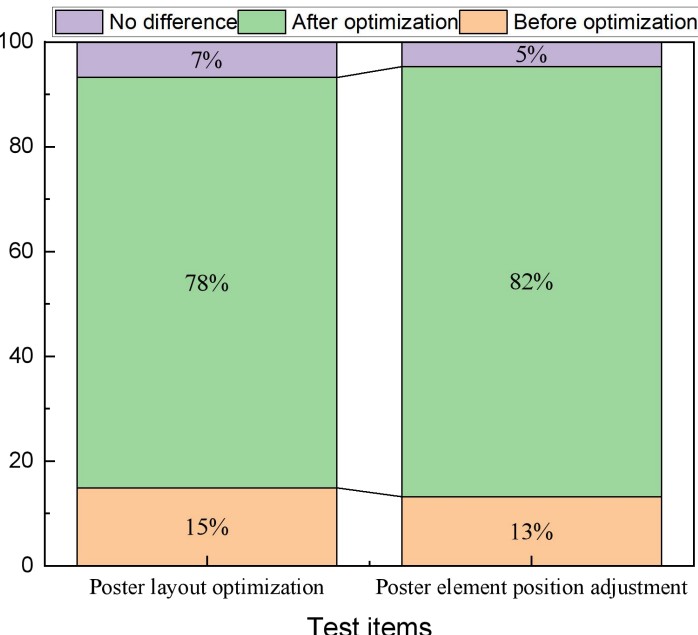

**Figure 6** **The user evaluation before and after the model optimization.** For both types of tests, more than 70% of the volunteers think the adjusted posters have better visual communication, while those who think no adjustment is better or there is no difference between the two.

is a great improvement, which shows that it is necessary to optimize the overall layout of posters by using AI technology.

## DISCUSSION

In terms of target detection, deep learning algorithms can be classified into two primary categories: Two Stage and One Stage detection algorithms. The former generates a sequence of candidate frames, followed by classifying samples using convolutional neural networks. In contrast, the latter transforms the target frame localization problem into a regression problem without generating candidates. Some of the commonly used algorithms in the two-stage approach are the region-convolutional neural network (R-CNN), the Fast-region-convolutional neural network (Fast-RCNN), and the region-based fully convolutional network (R-FCN) (*Kim, Sung & Park, 2020*). In view of the unique characteristics of target detection, the main objective of this study is to propose a method that leverages AI to design the layout. The Faster-RCNN model plays a pivotal role in this approach, as it uses RPN to learn and classify the elements of the format, thereby achieving recognition and positioning of the elements of the format. Specifically, it is utilized to learn the chosen poster case format and form the initial format template. The classification of the layout's constituent elements is primarily based on the previous structural annotation of the elements. However, its focus is on identifying the primary body and various text elements in the classified layout (*Soto & Yoo, 2019*). Once the classification of layout elements is completed, the position of the elements can be identified and located. This enables learning of the layout of the target case and completion of the learning migration of the layout. A comparison of the results in 'Experiment Results and Analysis' reveals that the use of Faster-R-CNN results in significant improvements in the Precision, Recall, and F1 score indexes. Moreover, in the subsequent optimization of the poster model, the evaluators' feedback underscores the significance of layout optimization. Thus, in the future, the incorporation of AI technology and the use of automatic layout optimization methods in the cultural sign design process can substantially enhance efficiency and satisfaction levels.

When designing public culture logos, it is essential to create a logo that is not only distinctive and eye-catching but also meets the requirements of visual communication. Apart from the logo itself, the optimal arrangement of related elements such as the theme, introduction, and others is crucial, necessitating designers to optimize the layout. The formal aesthetic law of layout represents an abstract generalization of the aesthetic law of layout design, which resonates with the audience aesthetically through contrast, proportion, white space, and other factors. The matching model based on the formal aesthetic law of layout is a more advanced abstract matching model that requires parameterization by the aesthetic law, owing to the current limitations of artificial intelligence technology (*Yang & Hsu, 2021*). In view of the visual communication requirements of public signs, this article proposes a poster learning and generation system based on the Faster-R-CNN framework, which can monitor various poster elements according to the requirements and generate optimized poster templates for designers to choose and match. This approach significantly enhances the design efficiency. The combination of AI and art design represents a major

trend, and with the development of quantum computers, the problem of unstructured design data computing and design uncertainty may be directly resolved, leading to a breakthrough from intelligent design to design intelligence in the future.

## CONCLUSION

This article proposes a framework for layout optimization and generation of cultural public representations, utilizing Faster-R-CNN in the context of current artificial intelligence. The framework initially performs element detection for cultural logo posters, with detection accuracies of 96.3%, 94.7%, and 92.8% for the main body, title, and introduction, respectively. In the subsequent practical test, the satisfaction rate of volunteers for the optimized logo exceeded 75%, providing new ideas for the design optimization of public cultural logos in the future. However, the article focuses only on simpler three-element poster logos and fails to address the generation of more complex cultural logos.

In future research, in addition to addressing the technical constraints of artificial intelligence, it is crucial to classify and label design elements more carefully and effectively, learn poster layouts composed of complex elements, and enhance the efficiency and quality of the intelligent layout design model. To achieve these objectives, it is recommended to introduce more diverse network models and expand the data range to generate a more robust model.

### Funding
The authors received no funding for this work.

### Competing Interests
The authors declare there are no competing interests.

### Author Contributions
- Xiang Gong conceived and designed the experiments, performed the experiments, analyzed the data, performed the computation work, prepared figures and/or tables, authored or reviewed drafts of the article, and approved the final draft.
- Jingyi Fang conceived and designed the experiments, performed the experiments, analyzed the data, authored or reviewed drafts of the article, and approved the final draft.

### Data Availability
The code is available in the Supplemental Files and at Zenodo: Public. (2023). Faster RCNN. Zenodo. https://doi.org/10.5281/zenodo.7864042.

The data is available at the South Australian Government Data Directory: https://data.sa.gov.au/data/dataset/stop-signs-for-state-maintained-roads.

The dataset comes from https://research.google.com/youtube8m/index.html (Names: YouTube-8M Segments Dataset; Institutions: YouTube-8M Users Group).

## Supplemental Information

Supplemental information for this article can be found online at http://dx.doi.org/10.7717/peerj-cs.1399#supplemental-information.

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
