# Peer review of "Design of public cultural sign based on Faster-R-CNN and its application in urban visual communication"

_PeerJ Computer Science, doi:10.7717/peerj-cs.1399_

## Round 0.1 · original submission · Minor Revisions

Dear Authors,

Please address the comments mentioned by the reviewers carefully and resubmit.

·

Basic reporting

Aiming at the visual communication optimization of cultural public representation under the current artificial intelligence background, this paper proposes a format optimization and generation framework based on Faster-R-CNN. The framework first completed the element detection of cultural identity posters, and the detection accuracy of the main body, title and introduction was 96.3%, 94.7% and 92.8%, respectively. In the follow-up practical test, the satisfaction rate of recruited volunteers to the optimized logo exceeded 75%, which provided a new idea for the design optimization of public cultural logo in the future. In order to be successfully accepted by this journal, the article still needs to be improved in the following aspects

(1) I'm unsure of the contribution of this work overall - the motivation of the article should be constructed according to this comment. Please make it clear;
(2) Some general statements should appear after sufficient arguments have been made, “the research content is relatively scattered, and the research on signs in the cultural field is less than that in other fields”;
(3) "object detection" is inappropriate as a keyword, and "Faster-R-CNN" may be better;
(4) In the introduction, I did not see enough advantages of artificial intelligence technology for visual communication typographic design;
(5) The references for Section 2.2 Target Detection and Automatic Layout Design, are too old, which should be replaced with some of the latest research, including improvements to GAN, etc;
(6) Add more introduction about Faster-R-CNN to reflect its recognition process, segmentation process and operation algorithm as much as possible;
(7) In the discussion part, I cannot see enough comparison of the outcomes of the present research with the previous related studies.

Experimental design

NO COMMENT

Validity of the findings

NO COMMENT

Additional comments

NO COMMENT

Reviewer 2 ·

Basic reporting

I believe the grammar is mostly clear and unambiguous except few minor revisions. Literature and background seems to be sufficient. The figures and tables are also depicted appropriately. Moreover, the results are self-contained.

Experimental design

The research seems to be in good scope of the journal. Research questions are largely meaningful and related. Rigorous analysis is performed, and methods well described.

Validity of the findings

The conclusion is representative of the whole manuscript and address the problem and methodology concisely. For future scope, please see the additional comments.

Additional comments

According to the visual communication needs of public cultural signs, this paper proposes an automatic layout optimization framework. In this framework, the model uses Faster-R-CNN to complete the detection and extraction of main elements in the logo poster. Then, the method of thirds in design is used to optimize the layout, so as to ensure that the cultural logo design conforms to the rules of visual communication. In the actual test, the average satisfaction rate of the cultural logo after layout optimization is more than 70%. This framework provides a new idea for the cultural logo design under the background of artificial intelligence in the future. In today's highly developed artificial intelligence, the author makes it possible to use it to optimize the cultural logo design. But the following issues need to be addressed:
(1) In the abstract, more attention is paid to the introduction of technical means rather than the research background;
(2) Some descriptions need to be replaced, such as "at home and abroad", please ensure the language style is consistent;
(3) Lines 38 and 41 repeat the semantic description;
(4) How does the author transform the design logic and computational logic to transform the design problem into a goal matching problem? There are currently no detailed instructions to resolve him;
(5) Author uses the current mainstream target detection algorithm Faster-R-CNN for image segmentation, and the innovation points of this paper are not clear.
(6) The title of Section 3.1 is not general, it needs to be changed;
(7) The formulas for Precision, Recall and F1score need not be introduced;
(8) The selection of model parameters should be reflected in the form of tables, and more details of the model should be reflected;
(9) Future scope of the research needs to be included and be more specific in the conclusion part.

---

## Round 0.2 · accepted · Accept

You paper is revised according to comments of the reviewers. Thank you for your contribution and congratulations!